# Evaluation of Implants with Different Macrostructures in Type I Bone—Pre-Clinical Study in Rabbits

**DOI:** 10.3390/ma13071521

**Published:** 2020-03-26

**Authors:** Amanda de Carvalho Silva Leocádio, Matusalém Silva Júnior, Guilherme José Pimentel Lopes de Oliveira, Gustavo da Col Santos Pinto, Rafael Silveira Faeda, Luis Eduardo Marques Padovan, Élcio Marcantonio Júnior

**Affiliations:** 1Department of Diagnosis and Surgery, School of Dentistry at Araraquara, Sao Paulo State University (UNESP), Araraquara 14801-385, Brazil; gustavo.dcsp@gmail.com; 2Post Graduation Course in Implantology, Latin American Institute of Dental Research (ILAPEO), Curitiba 80710-150, Brazil; jr@matusaodontologia.com (M.S.J.); padovan@iocp.com.br (L.E.M.P.); 3Department of Periodontology, School of Dentistry at Uberlândia, Federal University of Uberlândia, Uberlândia 38408-160, Brazil; guioliveiraodonto@hotmail.com; 4Post Graduation Course in Odontology, University of Araraquara/UNIARA, Araraquara 14801-320, Brazil; rafaelfaeda@gmail.com

**Keywords:** dental implants, macrostructure, osseointegration

## Abstract

The objective of this study was to assess the primary stability and the osseointegration process in implants with different macrostructures (Cylindrical vs. Hybrid Conical) in rabbit tibiae. Twenty-four (24) rabbits were used, divided into 3 experimental periods (2, 4 and 8 weeks) with 8 animals each. Each animal bilaterally received 2 implants from each group in the tibial metaphysis: Cylindrical Implant (CI) and Hybrid Conical Implant (HCI). All implants were assessed for insertion torque. After the experimental periods, one of the implants in each group was submitted to the removal counter-torque test and descriptive histological analysis while the other implant was used for microtomographic and histometric analysis (%Bone-Implant Contact). HCI implants showed higher insertion torque (32.93 ± 10.61 Ncm vs. 27.99 ± 7.80 Ncm) and higher % of bone-implant contact in the 8-week period (79.08 ± 11.31% vs. 59.72 ± 11.29%) than CI implants. However, CI implants showed higher values of removal counter-torque than HCI implants in the 8-week period (91.05 ± 9.32 Ncm vs. 68.62 ± 13.70 Ncm). There were no differences between groups regarding microtomographic data. It can be concluded that HCI implants showed greater insertion torque and bone-implant contact in relation to CI implants in the period of 8 weeks when installed in cortical bone of rabbits.

## 1. Introduction

The use of osseointegrated dental implants in the treatment of partial or total edentulism has been a widely used procedure in recent years [1,2]. However, despite the high rates of the clinical survival of contemporary dental implant systems, failures still occur, which may be associated with several mechanical or biological factors [3]. The lack of adequate bone formation or bone support volume to facilitate osseointegration has been reported as the main influence on the predictability and failure of the implant [4,5]. Survival rates also vary with the location of the implant in the oral cavity and other factors, such as implant design, biocompatibility, loading, bone density, surgical technique, and others [5,6].

In this context, a series of changes in the implant design and surgical techniques have been introduced to improve bone-implant contact, anchorage and stress distribution [7,8,9]. Many of these changes have led to successful implant therapy, even in the most difficult clinical situations. Because of this, the search for protocols that accelerate the osseointegration process, and consequently the prosthetic loading, has been the focus of several research papers in the area of Implantology [1,2].

Therefore, these structural modifications of the implants can be performed at the nano- or micro-structure level [10,11,12], and they would alter different stages in the stability of the implants, as changes in the macrostructure have been related to the achievement of the primary stability of the implants, while the changes in microstructure would affect the osseointegration process and, consequently, the secondary stability [11,12,13,14].

Considering that changes in the implant macrostructure have been reported as important for obtaining primary stability of implants in areas with poor quality bone, such as in the posterior region of the maxilla [15], studies with conical implants and threaded implants with no sharp edges and with healing chambers have been shown not only to improve primary stability but also to accelerate the osseointegration process [11]. In this context, the emergence of implants with a conical structure and with changes in the conformation of the threads, in order to make them more compressive, allows the installation of implants with good primary stability [16,17,18], as well as the use of implants with smaller sizes, since they can be used in areas with limited bone availability [19].

Consequently, that the macrostructure is an extremely important characteristic for obtaining primary stability, a parameter related to the success of the implant osseointegration, changes in the implant macrostructure have been proposed in order to improve the initial locking of the implants. Thus, the objective of this study was to verify the influence of an experimental implant with perforating compacting macrostructure (HCI—Hybrid Conical Implant: Helix Acqua, Neodent^®^, Grand Morse, Curitiba, Brazil) compared to a control implant (CI—Cylindrical Implant: Titamax Acqua, Neodent^®^, Cone Morse, Curitiba, Brazil) in the process of osseointegration in cortical bone (tibial metaphysis) of rabbits.

## 2. Materials and Methods

This project was carried out in accordance with the Ethical Principles for Animal Experimentation after approval by the Ethics Committee on Animal Use (ECAU) of the Faculty of Dentistry of Araraquara (FOAr-UNESP) (11/2016). For the present research, 24 male New Zealand albino rabbits (~5 months old and 4–5 kg) were used. The animals were kept in an environment with a temperature of 22–24 °C, with a controlled light cycle (12 h light and 12 h dark) and consumption of solid food and water ad libitum throughout the experimental period. The study was conducted according to the ARRIVE protocol.

### 2.1. Experimental Outline

The 24 rabbits were randomly divided into 3 experimental periods (2, 4 and 8 weeks). Each animal bilaterally received two implants from each group in the cortical bone of the animals’ tibia, with the side selected at random. Two different macrostructures were assessed: Hybrid Conical Implant (HCI) and Cylindrical Implant (CI).

The CI implant used in this study is characterized by a cervical diameter equal to the diameter of the implant body (3.75 mm in diameter × 11 mm in height). Presence of triangular and double threads that facilitate the quick implant insertion, with minimal trauma, and apex morphology with the presence of self-cutting chambers (Figure 1A,C,E). The HCI implant is characterized by an increased cervical diameter in relation to the implant body (3.75 mm in diameter × 11.5 mm in height). In addition, these implants have compacting and double trapezoidal threads, with a conical apex containing helical chambers designed to optimize secondary stability (Figure 1B,D,F).

### 2.2. Surgical Procedure

The animals were initially weighed and anesthetized intramuscularly, with a combination of ketamine (Quetamina Agener^®^, Agener União Ltd.a, São Paulo, SP, Brazil—0.35 mg/kg) and xylazine (Rompum, Bayer AS, São Paulo, SP, Brazil—0.5 mg/kg). Subsequently, trichotomy and antisepsis were performed with 10% iodinated polyvinylpyrrolidone (IPVP) with 1% active iodine in the right and left portions of the tibial metaphysis of the animal (Figure 2A). Local anesthesia (Mepivacaine Hydrochloride 2% + Adrenaline 1:100,000) was also applied in the region, to allow peripheral vasoconstriction, reducing local bleeding and optimizing the surgical procedure (Figure 2B). Then, using a No. 15 scalpel blade, a dermo-periosteal incision of approximately 5 cm in length was performed (Figure 2C,D). This allowed for a delicate dissection so that the bone surface of the tibial metaphysis was exposed, and the surgical beds were prepared according to the manufacturers’ recommendations (Neodent^®^—Curitiba, Brazil), using metal drills under abundant refrigeration with sterile saline solution (Figure 2E,F). Two implants of each type were installed in the tibiae, and the side was selected at random. The upper implants had a distance of 3 cm in relation to the lower ones (Figure 2G–I). Both implants had the same type of surface (Sandblasting + acid attack + immersion in 0.9% sodium chloride isotonic solution) and were manufacture with the same titanium alloy (Ti-6Al-4V). The cover screws were installed (Figure 2J) and then the soft tissues were repositioned and sutured plane by plane with resorbable thread (Vicryl^®^, ETHICON, Sao Paulo, Brazil) and non-resorbable thread (Shalon^®^-Nylon 3-0, Shalon surgical wires Ltda., Goias, Brazil) (Figure 2K,L). After surgery, all animals received a single intramuscular dose of antibiotic (Pentabiótico^®^, Wyeth-Whitehall Ltd.a, São Paulo, Brazil—0.1 mL/kg) and analgesic (Tramadol Hydrochloride 50 mg/mL, Tramadol^®^, Medley, São Paulo, Brazil—5 mg/kg IM). After the periods of 2, 4 and 8 weeks, the animals were euthanized through anesthetic overdose.

### 2.3. Biomechanical Assessment (Insertion Torque and Removal Torque)

All installed implants were submitted to the assessment of the insertion torque, and for that, the torque was noted after the installation of the implants (at the bone level). After euthanasia, in each analysis period (2, 4 and 8 weeks), the medial portions of the samples obtained from the tibia were reopened to expose the implants and perform the reverse torque. The samples were stabilized in a small vise, and a hexagonal wrench was connected to both the implant and the torque wrench (Tohnichi, model ATG24CN-S, Tokyo, Japan-with a graduated scale of 0.05 Ncm, measuring the strength from 3 to 24 Ncm), and an anti-clockwise movement was performed to remove the implants, increasing the torque until the rotation of the implant occurred inside the bone tissue, completely disrupting the bone-implant interface, when the torque wrench registered the maximum torque peak necessary for this disruption. This maximum peak required to move the implant was noted as the removal torque value. The remnants of the tibia related to the upper implants that were previously removed for the removal torque analysis were then reduced and immersed in 10% formaldehyde for 48 h, washed in running water for 4 h and then decalcified in Ethylenediaminetetraacetic Acid (E.D.T.A. 7%) for two months.

### 2.4. Descriptive Histology of Decalcified Sections

After the decalcification period, the parts from the sites where the implants were removed for analysis of the removal torque were embedded in paraffin, cut into a microtome (6 µm-thick) and stained using the hematoxylin-eosin (HE) technique. Five slides were obtained with 3 sections each in the central region of the site where the implant was inserted. Three sections were assessed that were 36 µm apart, and the first section for assessment was selected randomly. The sections were assessed by means of a DIASTAR optical microscope (Leica Reichert & Jung products, Wetzlar, Germany) with 5× and 10× magnifications, and the quality of bone tissue in the vicinity of the implant bed was assessed. This analysis was performed by an experienced trained examiner, blind for the type of implant used.

### 2.5. Microtomographic Analysis (µCT)

The parts from the sites where the implants were kept (lower tibial implants), after the fixation process with 4% formaldehyde for 48 h and washing with running water for 4 h, were dehydrated in an alcohol solution. Subsequently, they were subjected to the scanning process through the μCT and the following parameters were used: the size of the pixel image was 2000 × 1336 (18 μm), the thickness of the sections was 12 μm, the magnification of the image was 10×, the voltage of the X-ray tube was 50 kV, the beam was 496 μA, and the electrical current was adjusted to 0.1 mA. The three-dimensional images were reconstructed using a reconstruction software (NRecon 1.6.1.5, SkyScan N. V., Belgium). The parameters for reconstruction were: Beam Hardening Correction = 4%, Ring Artifact Correction = 3, Smoothing = 1, Postalignment = 1.00. Subsequently, the scanned images were reoriented in three planes (coronal, axial and sagittal) to standardize the position before performing the volumetric analysis. The measurements for the Volumetric analysis (3D) were carried out using specific software (CT Analyzer 1.10.1.0, SkyScan N. V., Belgium), following the selection of an area of interest (ROI—region of interest) in a cylindrical shape that circumscribed in 0.5 cm the diameter of the implants. Although the implants received a cover screw, there was bone formation inside the cover screw in some cases. In order to prevent this bone formation from interfering with the analysis of the volume of mineralized tissue around the implant, a second ROI was established to remove the volume of mineralized tissues that could have been formed in this region. With the results obtained in the two ROI’s, it was possible to define the volume of mineralized tissues using the following formula:

Total bone volume of mineralized tissues in the main ROI—Volume of mineralized tissues within the cover screw = Volume of mineralized tissues (BV/TV).

The grayscale threshold used was 25–90, and the values of the volume of mineralized tissue around the implant were obtained as a percentage. The entire analysis was performed by a single trained examiner, who was blind for the type of training performed [20].

### 2.6. Histometric Analysis (BIC)

The biopsies with the lower implants that were submitted to the fixation process and microtomographic analysis were subsequently dehydrated in an alcohol solution in a series of increasing concentrations. The plastic infiltration was performed with mixtures of glycolmethacrylate (Technovit 7200 VLC) and ethyl alcohol, following gradual variations, ending with two infiltrations of pure glycol methacrylate. After plastic infiltration, the specimens were embedded in resin and polymerized. Therefore, the specimens were sectioned longitudinally, along the main axis of the implant, using a high precision diamond disk. The blocks were mounted on an acrylic slide with the help of Tecnovit 4000 resin (Kulzer, Wehrheim, Germany). Using a cutting and micro-wear system (Exact-Cutting, System, Apparatebau Gmbh, Hamburg, Germany), the blades were processed with a section of approximately 50–70 μm thick. The pieces were stained with Stevenel’s Blue for histomorphometric analysis. Histometry assessed the percentage of mineralized bone in direct contact with the implant surface (BIC—bone to implant contact) in the extension of the cervical third of the implant. The measurements were made using a DIASTAR optical microscope (Leica Reichert & Jung products, Wetzlar, Germany), with a 10-fold magnification objective lens, through which the images were captured and sent to a PC, with the aid of a video camera (Leica Reichert & Jung products, Wetzlar, Germany). The values were determined using image analysis software (Image J, Jandel Scientific, San Rafael, CA, USA), by a blind examiner, calibrated and trained for this analysis [21].

%BIC = Direct contact of the bone with the implant surface × 100/Extension of the cervical third of the implant.

### 2.7. Statistical Analysis

The data from the biomechanical (insertion and removal torque), microtomographic (BV/TV) and histometric (%BIC) analyses were submitted to the Shapiro-Wilk normality test which confirmed that the data were distributed according to the central distribution theorem. The parametric paired t-test was used for the inferential analysis of the data comparing the different groups of macrostructures (CI vs. HCI). The One-way ANOVA test was applied to compare the different assessment periods within each group. The GraphPad Prism 6 software (San Diego, CA, USA) was employed in the statistical tests that were applied at the significance level of 5%. Sample calculation was performed using the paired t-test based on the histometric data of bone-implant contact from the study by Faeda et al. 2012 [21], which assessed the effect of different implant surfaces on osseointegration in rabbits. It was found that the difference between the BIC averages among different implant surfaces in order to promote a statistically significant difference was 25.95% (SD = 8.34). Therefore, the use of 8 rabbits per group in each period would be sufficient to obtain a power β and α of the study greater than 0.9 and equal to 0.05, respectively.

## 3. Results

The animals tolerated well the surgical procedure and remained healthy throughout the experimental period.

### 3.1. Biomechanical Analysis

It was found that the HCI implants had a higher insertion torque than the CI implants. There was a progressive increase in implant removal torques in all groups, regardless of the type of macrostructure assessed. It was found that the CIs presented higher removal torque than HCIs in the period of 8 weeks. Figure 3 and Figure 4 and Table 1 show the data on the mean and standard deviation of the biomechanical analysis.

### 3.2. Microtomographic Analysis

There were no differences between groups regarding the BV/TV data. However, there was a progressive increase in this parameter with the increase in the time of the experimental periods in both groups. Figure 5 and Table 2 show the data on the mean and standard deviation from the BV/TV data obtained through microtomographic analysis of all groups. Figure 6 shows the microtomographic images of the implants with the different macrostructures.

### 3.3. Descriptive and Histometric Histological Analysis

The histological description was performed in decalcified sections in the region associated with the first thread of the implants. There were no differences in the histological aspects of bone tissue associated with the different macrostructures of the implants, thus, this description was performed together, varying only the description of the different experimental periods.

At two weeks, it was possible to observe the formation of new bone in the region of the threads that presented a trabecular aspect with rounded osteocytes, immature mineralized matrix without organization in the form of concentric lamellae, active osteoblasts with intense organization of the formation of Haversian channels and presence of medullary tissue. At 4 weeks, it was observed an increase in the formation of the mineralized matrix associated with a change in the appearance of the matrix to a more organized condition with the formation of concentric lamellae and a reduction in the number and diameter of the Haversian channels. There was also a reduction in medullary tissue, presence of rounded osteocytes in large numbers and a lower number of osteoblasts. The appearance of peri-implant bone tissue has changed little over the 8-week period compared to the 4-week period. The presence of a mineralized matrix was observed in an advanced stage of mineralization, with the presence of well-defined Haversian channels, presence of rounded osteocytes, organization of the mineralized matrix in concentric lamellae and reduced amount of medullary tissue (Figure 7).

It was found that HCI implants (79.08 ± 11.31%) had a higher %BIC in the 8-week period compared to CI implants (59.72 ± 11.29%). Figure 8 and Table 3 show the data of the mean and standard deviation from the %BIC data obtained through the histometric analysis of all groups. Figure 9 shows images representative of the non-decalcified sections of the implants with different macrostructures and in the different experimental periods.

## 4. Discussion

In general, in this study, it was found that HCI implants showed better primary stability, which consequently resulted in greater bone-implant contact at the end of the 8 weeks of assessment, which demonstrates that this type of macrostructure can clinically receive occlusal loads earlier, be used in bones of lower density and accelerate the osseointegration process when compared to CI implants.

It has been determined that the implant macrostructure promotes changes in terms of obtaining primary stability [22,23]. The results of this study support studies that demonstrated that conical implants have superior primary stability when compared to cylindrical implants. However, despite the statistically significant difference obtained in this study, cylindrical implants also showed good results for primary stability [18,24]. This finding may be related to the fact that the experiment was carried out on the tibial bone, which has the characteristics of a type I bone, due to the presence of a thick cortical region that allowed the implants to lock properly [25,26]. It is likely that, in a bone of poorer quality, the differences in primary stability for HCI implants are even greater when compared to CI implants, a hypothesis that needs to be tested in the future.

Conflicting data in this study was that the removal torque of CI implants was higher than that of HCI implants in the period of 8 weeks. A factor that could have interfered with this aspect would be that the higher insertion torque of the HCI implants would have induced greater necrosis of the cortical bone tissue and delayed the osseointegration process [27,28]. However, the insertion torque values of the HCI implants did not exceed 45 Ncm, which is not related to these adverse events [28]. In addition, the histological assessment of this study proved the similarity between implants with different macrostructures in the same experimental period assessed. The implants in this study had bicortical locking, where their apexes were locked to the posterior cortical bone of the tibia of the rabbits. Despite the fact that the surface area of the implants was not measured, it is possible to observe that the apexes of HCI implants are smaller than those of CI implants, it is possible that this increase in the contact area of the surface of the CI implants may have benefited the increased removal torque of these implants [29]. HCI implants are indicated for all types of bone densities. In the case of type I and II bones, an overrun drill should be used. However, during the installation of these implants, manual torque and counter-torque are often necessary to avoid damage to adjacent tissues. For this reason, the apex of HCI implants (Figure 1F) has already been developed with low intensity helical cameras to facilitate counter torque during manual installation. This fact can also explain the removal torque result of this study.

It has also been reported in the literature that primary stability is essential for the osseointegration process to occur in a predictable way and that the degree of osseointegration is correlated with the quality of this stability [30,31,32], a fact reinforced by the findings of this study that demonstrated a higher degree of %BIC associated with HCI implants when compared to what was observed in the CI implants in the 8-week period, however, without altering the quality of the newly formed bone. This finding demonstrates that the HCI macrostructure benefits the acceleration of the osseointegration process and that perhaps this difference is even more relevant in more challenging clinical conditions (e.g., low-density native bone, grafted areas), however, the animal model used does not allow to infer on the real impact of the HCI macrostructure on osseointegration in these clinical conditions.

The results of this study should be interpreted with caution because, in addition to the limitations mentioned above, this model cannot extrapolate clinical conditions such as the application of immediate or early loading, which could alter the osseointegration process in these implants. In addition, the bicortical locking offered by the tibia of rabbits, which in fact helps in the primary stability of implants, is not a common event to be expected in daily clinical practice, where implants usually lock in their most coronal portion in the cortical bone of the maxilla or the mandible. Thus, the effect of HCI on osseointegration still requires further investigation.

## 5. Conclusions

According to the results from and within the limits of this study, it is observed that HCI Implants showed greater insertion torque and bone-implant contact in relation to CI implants in the period of 8 weeks when installed in cortical bone of rabbits. However, the removal torque of CI implants was higher than that of HCI implants in the period of 8 weeks.

## Figures and Tables

**Figure 1 materials-13-01521-f001:**
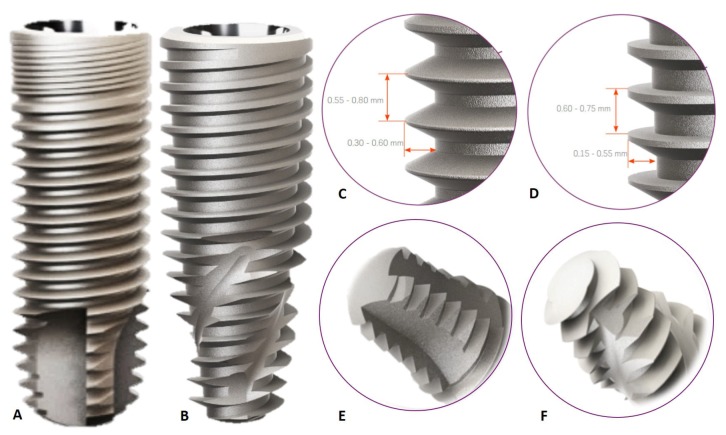
Macrostructure of implants installed in the cortical bone of rabbits. Body (**A**), threads (**C**) and apex (**E**) of the Cylindrical Implant; Body (**B**), threads (**D**) and apex (**F**) of the Hybrid Conical Implant.

**Figure 2 materials-13-01521-f002:**
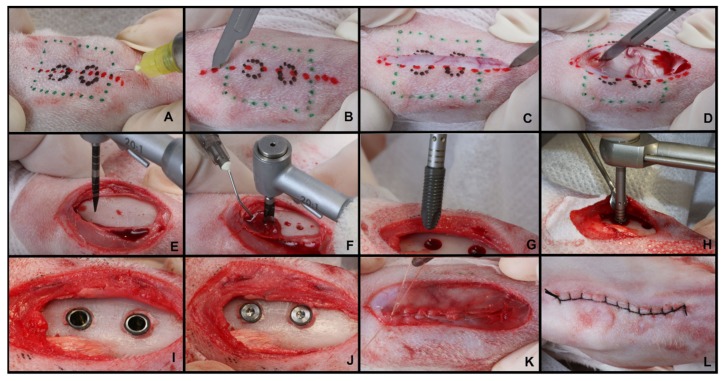
Surgical procedure for installing implants in the tibial metaphysis of rabbits. Schematic drawing to define the tibial metaphysis and local anesthesia (**A**); Incision in layers (**B**–**D**); Detachment and exposure of the tibia, and perforation according to the manufacturer’s recommendations under abundant sterile saline irrigation (**E**,**F**); Installation of the implants (**G**,**H**); Implants and cover screws installed (**I**,**J**); Internal suture of the muscular fascia with resorbable thread and external suture with non-resorbable thread (**K**,**L**).

**Figure 3 materials-13-01521-f003:**
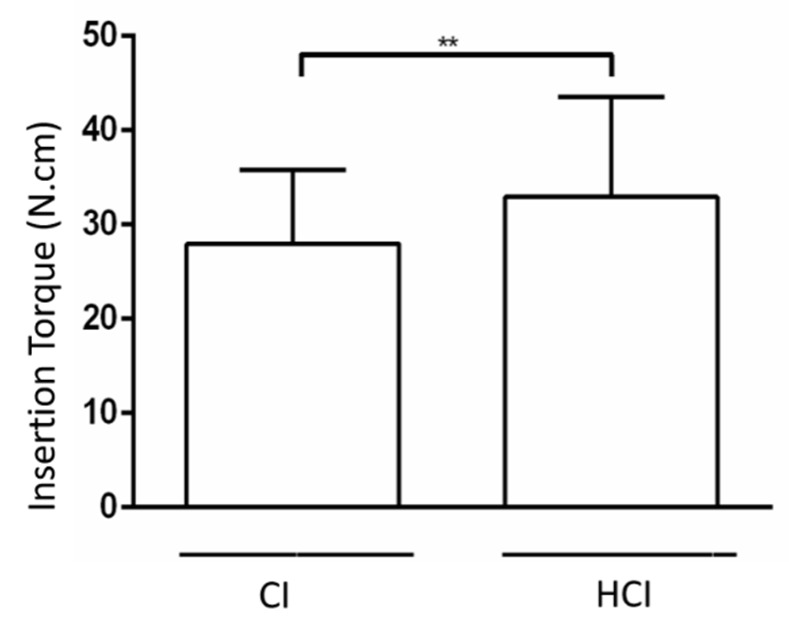
Mean and standard deviation of implant insertion torque with different macrostructures. ** *p* < 0.01—Differences between groups of implants with different macrostructures—Paired t-test.

**Figure 4 materials-13-01521-f004:**
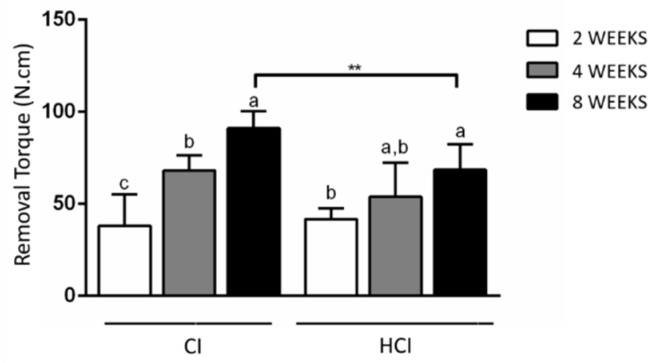
Mean and standard deviation of implant removal torque with different macrostructures. ** *p* < 0.01—Differences between groups of implants with different macrostructures—Paired t-test. Different letters represent different levels of statistically significant differences among the periods within each group (*p* < 0.05). One-way ANOVA complemented by the Tukey test.

**Figure 5 materials-13-01521-f005:**
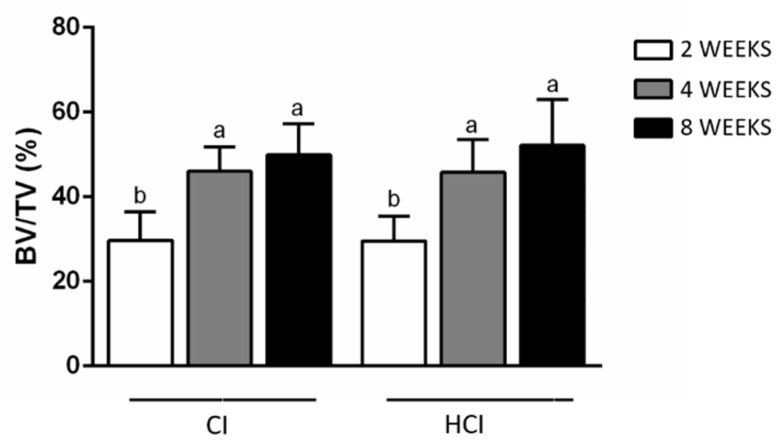
Mean and standard deviation for BV/TV data obtained through microtomographic analysis of all groups. Different letters represent different levels of statistically significant differences among the periods within each group (*p* < 0.05). One-way ANOVA complemented by the Tukey test.

**Figure 6 materials-13-01521-f006:**
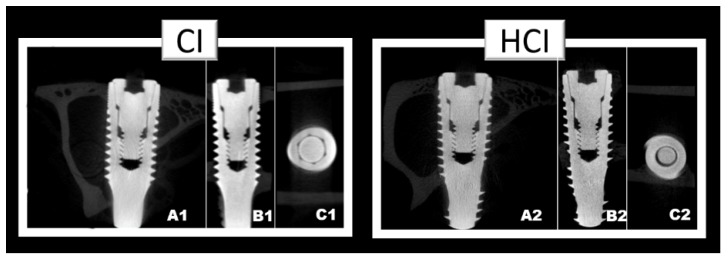
Microtomographic images representative of implants with different macrostructures CI: Cylindrical Implant (1) and HCI: hybrid conical implant (2). Axial slice (**A**); Sagittal slice (**B**); and Coronal slice (**C**).

**Figure 7 materials-13-01521-f007:**
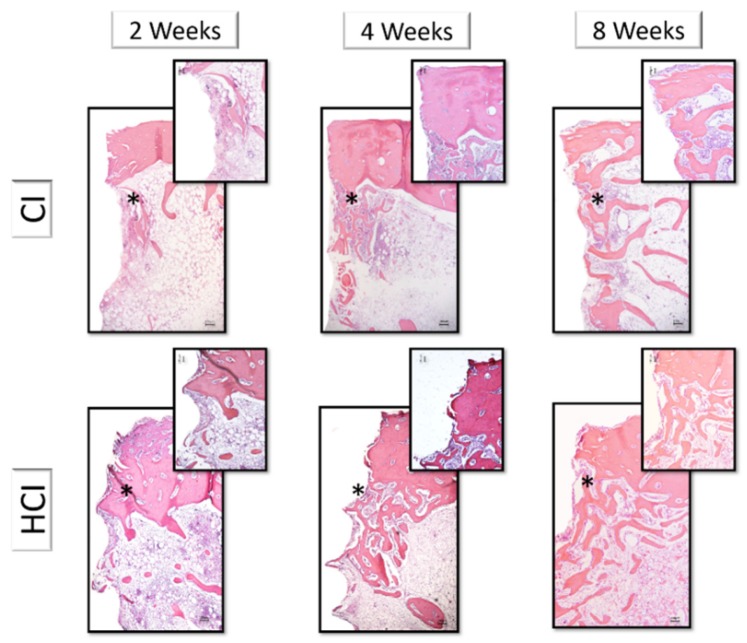
Histological images representative of decalcified sections of implants with different macrostructures and in different experimental periods. CI: Cylindrical Implant and HCI: Hybrid Conical Implant. Both in the periods of 2, 4 and 8 weeks. 5× and 10× *g*.

**Figure 8 materials-13-01521-f008:**
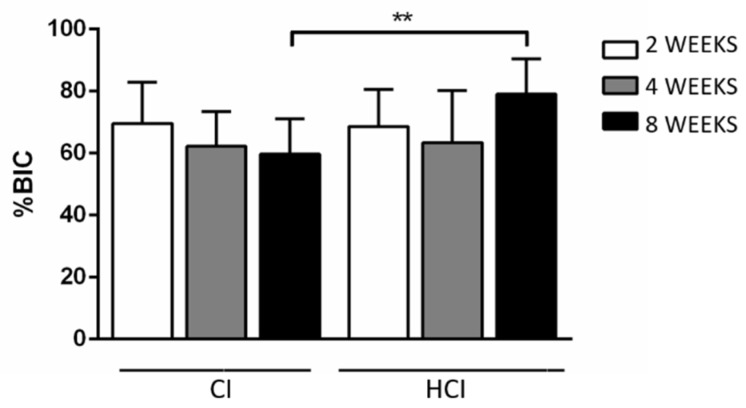
Mean and standard deviation for %BIC data obtained through histometric analysis of all groups. ** *p* < 0.01—Differences between groups of implants with different macrostructures—Paired t-test.

**Figure 9 materials-13-01521-f009:**
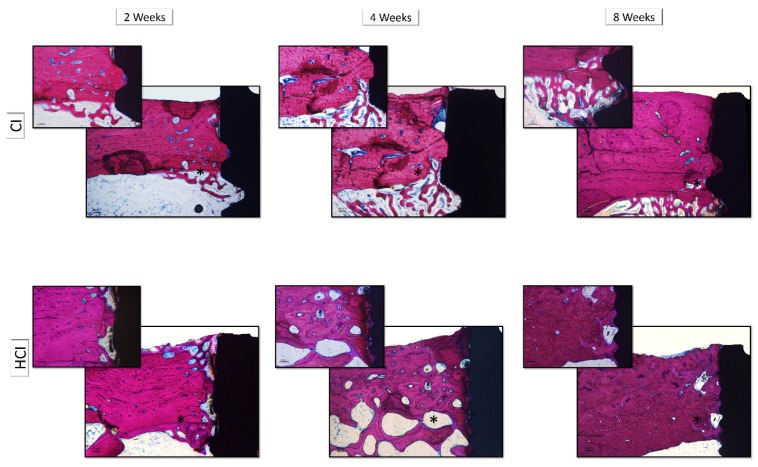
Histological images representative of the non-decalcified sections of the implants with different macrostructures and in different experimental periods. CI: Cylindrical Implant and HCI: Hybrid Conical Implant. Both in the periods of 2, 4 and 8 weeks. 5× and 10× *g*.

**Table 1 materials-13-01521-t001:** Data on the mean and standard deviation for the insertion and removal torque of all groups.

-	Insertion Torque	Removal Torque		
Implant type/Period	-	2 weeks	4 weeks	8 weeks
CI	27.99 ± 7.80 *	38.01 ± 17.09 ^c^	68.01 ± 8.46 ^b^	91.05 ± 9.32 **^,a^
HCI	32.93 ± 10.61 *	41.69 ± 5.98 ^b^	53.88 ± 18.50 ^a,b^	68.62 ± 13.70 **^,a^

* *p* < 0.01—Differences between groups of implants with different macrostructures—Paired t-test. ** *p* < 0.01—Differences between groups of implants with different macrostructures—Paired t-test. Different letters (^a^, ^b^ and ^c^) represent different levels of statistically significant differences among the periods within each group (*p* < 0.05). One-way ANOVA complemented by the Tukey test.

**Table 2 materials-13-01521-t002:** Data on the mean and standard deviation for BV/TV data obtained through microtomographic analysis of all groups.

Implant Type/Period	2 Weeks	4 Weeks	8 Weeks
CI	29.68 ± 6.77 ^b^	46.06 ± 5.68 ^a^	49.95 ± 7.36 ^a^
HCI	29.58 ± 5.78 ^b^	45.80 ± 7.78 ^a^	52.19 ± 10.77 ^a^

Different letters (^a^ and ^b^) represent different levels of statistically significant differences among the periods within each group (*p* < 0.05). One-way ANOVA complemented by the Tukey test.

**Table 3 materials-13-01521-t003:** Data on the mean and standard deviation for %BIC data obtained through histometric analysis of all groups.

Implant Type/Period	2 Weeks	4 Weeks	8 Weeks
CI	69.64 ± 13.22	62.21 ± 11.19	59.72 ± 11.29 *
HCI	68.62 ± 11.97	63.49 ± 16.77	79.08 ± 11.31 *

* *p* < 0.01—Differences between groups of implants with different macrostructures—Paired t-test.

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
