# Peer review of "Evaluation of Implants with Different Macrostructures in Type I Bone—Pre-Clinical Study in Rabbits"

_materials, 2020, doi:10.3390/ma13071521_

Round 1

Reviewer 1 Report

Article: Evaluation of implants with different macrostructures in type I bone - Pre-clinical study in rabbits.

The purpose of the paper is to evaluate the primary stability and osseointegration of two different designs of implants (cylindrical body versus conical body). It is correctly planned and executed, but it has an important defect: the Resonance Frequency Analysis is not measurable. As we know, the insertion torque only measures the force at the first time, not the final one, when the implant is stable. Furthermore, secondary stability should once again be measured better than histomorphometry, on the other hand, the examinations are forced to practice disengagement and evaluate that that torque desinsertation. It can be an important distortion

In fact, the RFA is related to primary and special secondary stability. Moreover, the author cannot conclude as they said: That the conical design shows more primary stability than cylindrical one, because it has not been determined during the examination. In addition, the author concluded that the cylindrical had better disinsertion torque, and that is an aspect that has not been shown in the paper. The conclusion is not relevant to the results shows. The cylindrical design has higher ISQ values? Has more secondary stability than the others?

The calculation of the sample size by groups and times should be specified.

Bibliography:

6) First author repeated twice.

8) mistake in the name of the magazine

11) mistake in the name of the magazine

23) DELETE is the 16 that has been repeated! Renumber in text and bibliography from 24 to 32.

32) missing Nº / vol… 9 (1)

Author Response

Response to Reviewer 1 Comments

Article: Evaluation of implants with different macrostructures in type I bone - Pre-clinical study in rabbits.

Point 1: The purpose of the paper is to evaluate the primary stability and osseointegration of two different designs of implants (cylindrical body versus conical body). It is correctly planned and executed, but it has an important defect: the Resonance Frequency Analysis is not measurable. As we know, the insertion torque only measures the force at the first time, not the final one, when the implant is stable. Furthermore, secondary stability should once again be measured better than histomorphometry, on the other hand, the examinations are forced to practice disengagement and evaluate that that torque desinsertation. It can be an important distortion

Response 1: The reviewer is right regarding the importance of the RFA analysis in the evaluation of the dental implant stability. The RFA in fact is a good method to evaluate the implant stability, however the insertion torque is an important parameter that the clinician uses to make important decision as the time of implant loading. So, despite the fact that we didn´t measured the final stability. The removal torque and histometric analysis together are classical and reasonable methods to evaluate the osseointegration and our main conclusions are based on these important methods.

Point 2: In fact, the RFA is related to primary and special secondary stability. Moreover, the author cannot conclude as they said: That the conical design shows more primary stability than cylindrical one, because it has not been determined during the examination. In addition, the author concluded that the cylindrical had better disinsertion torque, and that is an aspect that has not been shown in the paper. The conclusion is not relevant to the results shows. The cylindrical design has higher ISQ values? Has more secondary stability than the others?

Response 2: We agreed with the reviewer and we rewrite apart of the conclusion since the ISQ test were not performed. We also gave more importance about the fact that the cylindrical implants had higher removal torque values due than the conical implants. Our explanation about this issue were related at the discussion section. After the corrections, the new conclusion is: “According to the results from and within the limits of this study, it is observed that HCI Implants showed greater insertion torque and bone-implant contact in relation to CI implants in the period of 8 weeks when installed in cortical bone of rabbits. However, the removal torque of CI implants was higher than that of HCI implants in the period of 8 weeks.” (LINE 31 and 332)

Point 3: The calculation of the sample size by groups and times should be specified.

Response 3: The sample size detected by the sample size calculation was the same in each group and period. We rewrite this sentence in order to become this information more clear: ‘Sample calculation was performed using the paired t-test based on the histometric data of bone-implant contact from the study by Faeda et al. 2012 [20], which assessed the effect of different implant surfaces on osseointegration in rabbits. It was found that the difference between the BIC averages among different implant surfaces in order to promote a statistically significant difference was 25.95% (SD=8.34). Therefore, the use of 8 rabbits per group in each period would be sufficient to obtain a power β and α of the study greater than 0.9 and equal to 0.05, respectively” (LINE 205)

Point 4: Bibliography:

6) First author repeated twice.

Response 3: Bibliography:

6) Cameron HU, Pilliar RM, Macnab I. The effect of movement on the bonding of porous metal to bone. J Biom Mat Res. 1973; 7: 301-311.

Incorrect: 7) Steigenga JT, Steigenga JT, Al-Shammari KF, Nociti FH, Misch CE, Wang HL.  Dental implant design and its relationship to long-term implant success. Impl Dent. 2003; 12 :306-317.

Reference nº7 presented the mentioned repetition.

Correct: 7) Steigenga JT, Al-Shammari KF, Nociti FH, Misch CE, Wang HL.  Dental implant design and its relationship to long-term implant success. Impl Dent. 2003; 12 :306-317.

Reference 7 checked.  Line 371.

Response 3: Bibliography:

8) mistake in the name of the magazine:

Incorrect: Junker R, Dimakis A, Thoneick M, Jansen JA. Effects of implant surfasse coatings and composition on bone integration: A systematic review. Clinl O Impl Res. 2009; 20: 185-206.

Correct: Junker R, Dimakis A, Thoneick M, Jansen JA. Effects of implant surfasse coatings and composition on bone integration: A systematic review. Clin Oral Implants Res. 2009; 20: 185-206.

Reference 8 checked. Line 373.

Response 3: Bibliography:

11) mistake in the name of the magazine

Incorrect: Javed F, Romanos GE. The role of primary stability for successful immediate loading of dental implants. A literature review. J of Dentistry. 2010; 38(8): 612-620.

Correct: Javed F, Romanos GE. The role of primary stability for successful immediate loading of dental implants. A literature review. J. Dent. 2010; 38(8): 612-620.

Reference 11 checked. Line 379.

23) DELETE is the 16 that has been repeated! Renumber in text and bibliography from 24 to 32.

Reference 23 deleted. And the others renumbered.

32) missing Nº / vol… 9 (1)

Reference 32 now has been changed to 31.

Incorrect: 32.   Bataineh AB, Al-Dakes AM. The influence of length of implant on primary stability: An in vitro study using resonance frequency analysis. J Clin Exp Dent. 2017;9: e1 – e6.

Correct: 31.     Bataineh AB, Al-Dakes AM. The influence of length of implant on primary stability: An in vitro study using resonance frequency analysis. J Clin Exp Dent. 2017; 9(1): e1–e6.

Reference 31 checked. Line 430.

Reviewer 2 Report

I enjoyed this well-performed manuscript. The experiments have been carefully designed and performed and the results are convincing. I just wish to put the authors’ attention on some points, as listed below.

1) Results: perhaps repetition of the experimental values and SEM in the text is unnecessary, considering that the same values have been clearly reported in the tables.

2) Results, microtomographic analysis: the measured parameter BV/TV should be spelled out and explained at its first mention: in fact, this information is missing in the methods.

3) Charts 2 and 3: please define the p values corresponding to the letters a and b.

4) Tables 1, 2 & 3: please add a further bottom line with the p values of the statistical comparisons between CI and HCI at the various time points.

5) Figure 4: the insets at a slightly higher magnification add little additional information: these could be replaced with others at even higher magnification to better depict the cellular details described in the text. Are x5 and x10 the values of the final magnifications or of the used objectives?

6) Figure 5: also in this case the insets are poorly informative, being almost similar to the larger panels, and could be removed: this would allow to enlarge the remaining panels within the same figure area.

7) Discussion: a possible explanation for the lower removal torque of HCI implants could be that their overall surface interlocked with bone is lower than that of CI because of their different 3-D geometry. Could the authors perform an estimate (or have data provided by the supplier) of the whole intra-bony surface area of the two types of implants?

Author Response

Response to Reviewer 2 Comments

Article: Evaluation of implants with different macrostructures in type I bone - Pre-clinical study in rabbits.

I enjoyed this well-performed manuscript. The experiments have been carefully designed and performed and the results are convincing. I just wish to put the authors’ attention on some points, as listed below.

Point 1:

1) Results: perhaps repetition of the experimental values and SEM in the text is unnecessary, considering that the same values have been clearly reported in the tables.

Response 1:

We remove the values ​​from the text and keep the tables.

LINE 210

“Biomechanical Analysis

It was found that the HCI implants had a higher insertion torque than the CI implants. There was a progressive increase in implant removal torques in all groups, regardless of the type of macrostructure assessed. It was found that the CIs presented higher removal torque than HCIs in the period of 8 weeks. Charts 1 and 2 and Table 1 show the data on the mean and standard deviation of the biomechanical analysis.

Microtomographic Analysis

There were no differences between groups regarding the BV/TV data. However, there was a progressive increase in this parameter with the increase in the time of the experimental periods in both groups. Charts 3 and Table 2 show the data on the mean and standard deviation from the BV/TV data obtained through microtomographic analysis of all groups. Figure 3 shows the microtomographic images of the implants with the different macrostructures.

Descriptive and histometric histological analysis. It was found that HCI implants had a higher %BIC in the 8-week period compared to CI implants. Table 3 shows the data of the mean and standard deviation from the %BIC data obtained through the histometric analysis of all groups. Figure 5 shows images representative of the non-decalcified sections of the implants with different macrostructures and in the different experimental periods.”

Point 2:

2) Results, microtomographic analysis: the measured parameter BV/TV should be spelled out and explained at its first mention: in fact, this information is missing in the methods.

Response 2:

We spelled out and explained the mean of the BV/TV at the material and methods section: ´ In order to prevent this bone formation from interfering with the analysis of the volume of mineralized tissue around the implant, a second ROI was established to remove the volume of mineralized tissues that could have been formed in this region. With the results obtained in the two ROI’s, it was possible to define the volume of mineralized tissues using the following formula:

            Total bone volume of mineralized tissues in the main ROI – Volume of mineralized tissues within the cover screw = Volume of mineralized tissues (BV/TV). (LINE 168)

            The grayscale threshold used was 25-90, and the values of the volume of mineralized tissue around the implant were obtained as a percentage. The entire analysis was performed by a single trained examiner, who was blind for the type of training performed [19].”

Point 3:

3) Charts 2 and 3: please define the p values corresponding to the letters a and b.

Response 3:

We added the definition of the p values at the legends of the charts 2 and 3: “ Chart 2: Mean and standard deviation of implant removal torque with different macrostructures. **p<0.01- Differences between groups of implants with different macrostructures - Paired t-test. Different letters represent different levels of statistically significant differences among the periods within each group (p<0.05). One-way ANOVA complemented by the Tukey test. (LINE 222)

Chart 3: Mean and standard deviation for BV/TV data obtained through microtomographic analysis of all groups.

Different letters represent different levels of statistically significant differences among the periods within each group (p<0.05). One-way ANOVA complemented by the Tukey test.” (LINE 241)

Point 4:

4) Tables 1, 2 & 3: please add a further bottom line with the p values of the statistical comparisons between CI and HCI at the various time points.

Response 4:

We added theses information’s in all the tables. (LINE 224 / 243/ 277)

Point 5:

5) Figure 4: the insets at a slightly higher magnification add little additional information: these could be replaced with others at even higher magnification to better depict the cellular details described in the text. Are x5 and x10 the values of the final magnifications or of the used objectives?

Response 5: We agreed with the reviewer and we replace the images with images with higher quality. We also changed the information of the augmentation since the final augmentation was 50 and 100X, since the previous values was regarding the objectives lens used. (LINE 269)

Point 6:

6) Figure 5: also in this case the insets are poorly informative, being almost similar to the larger panels, and could be removed: this would allow to enlarge the remaining panels within the same figure area.

Response 6:

We replace the images according the reviewer concern. (LINE 280)

Point 7:

7) Discussion: a possible explanation for the lower removal torque of HCI implants could be that their overall surface interlocked with bone is lower than that of CI because of their different 3-D geometry. Could the authors perform an estimate (or have data provided by the supplier) of the whole intra-bony surface area of the two types of implants?

Response 7: Unfortunately, we don´t have access of this data. We discuss a lit bit this topic at the discussion section based only on our observations of the differences in the morphology of the implant’s apex tested. We rewrite a part of the discussion in order to clarify this issue and limitations.” The implants in this study had bicortical locking, where their apexes were locked to the posterior cortical bone of the tibia of the rabbits. Despite the fact that the surface area of the implants were not measured, it is possible to observe that the apexes of HCI implants are smaller than those of CI implants, it is possible that this increase in the contact area of the surface of the CI implants may have benefited the increased removal torque of these implants [28].” (LINE 306)

Round 2

Reviewer 1 Report

Ok

corrections have been made. In a future study they should also use RFA